# Characteristics of Individuals with Disagreement between Home and Ambulatory Blood Pressure Measurements for the Diagnosis of Hypertension

**DOI:** 10.3390/healthcare8040457

**Published:** 2020-11-03

**Authors:** Chee Hae Kim, Je Sang Kim, Moo-Yong Rhee

**Affiliations:** 1Cardiovascular Center, Dongguk University Ilsan Hospital, Goyang 10326, Korea; chhkim@dumc.or.kr (C.H.K.); drjesang@dumc.or.kr (J.S.K.); 2College of Medicine, Dongguk University, Gyeongju-si, Gyeongbuk 38066, Korea

**Keywords:** hypertension, out-of-office blood pressure, home blood pressure measurements, ambulatory blood pressure measurements, disagreement

## Abstract

Home and ambulatory blood pressure (BP) measurements are recommended for the diagnosis of hypertension. However, the clinical characteristics of individuals showing a diagnostic disagreement between their home and ambulatory BP measurements are unclear. Of the 470 individuals who were not on antihypertensive drug treatment with a BP ≥140/90 mmHg at an outpatient clinic, 399 who had valid office, home, and ambulatory BP results were included. Hypertension was diagnosed based on an average home BP ≥135/85 mmHg and/or an average daytime ambulatory BP ≥135/85 mmHg. The participants were divided into three groups: Agree-NT (home and ambulatory BP normotension), Disagree (home BP normotension and ambulatory BP hypertension, or home BP normotension and ambulatory BP hypertension), and Agree-HT (home and ambulatory BP hypertension). Eighty-four individuals (21.1%) were classified as the Disagree group. The mean serum creatinine, triglycerides, and electrocardiogram voltage in the Disagree group were intermediate between those observed in the Agree-NT and the Agree-HT group. In the Disagree group, the mean levels of office and home diastolic BP, all of the components of ambulatory BP, the aortic systolic BP, and the BP variabilities were found to be intermediate between those of the Agree-NT and the Agree-HT groups. These results indicate that individuals showing a diagnostic disagreement between their home and ambulatory BP may have cardiovascular risks that are intermediate between those with sustained home and ambulatory normotension and hypertension.

## 1. Introduction

Accurate blood pressure (BP) measurements are essential for the diagnosis and treatment of hypertension [1]. Office BP measurements have been widely used to monitor BP because they are conveniently and rapidly obtained, but they are easily confounded by various factors [2]. Therefore, current guidelines recommend out-of-office blood pressure measurements for the optimal diagnosis and treatment of hypertension [1,3]. Both home and ambulatory BP measurements provide BP values based on multiple measurements, reflect BP variations, and have the ability to identify white-coat or masked hypertension, which are commonly misdiagnosed by using office BP only. Furthermore, home BP and ambulatory BP showed a better association with a future cardiovascular risk or target organ damage compared to office BP [4,5,6].

Although home and ambulatory BP measurements have similar advantages over office BP measurements, are evenly recommended to confirm the diagnosis of white-coat and masked hypertension in the current guidelines, and share the same threshold for hypertension (i.e., ≥135/85 mmHg for home BP and for the daytime ambulatory BP) [1,3], they can provide different BP values and result in contradictory diagnoses. However, the clinical characteristics of individuals that exhibit diagnostic disagreement between home BP and ambulatory BP measurements, and whether they should be diagnosed as normotension or hypertension, are still undetermined.

Accordingly, we aimed to identify the clinical characteristics of individuals related to diagnostic disagreement between home and ambulatory BP measurements.

## 2. Materials and Methods

### 2.1. Study Population

From November 2015 to November 2019, 470 individuals who were not on antihypertensive drug treatment with BP of ≥140/90 mmHg, measured by doctors at the outpatient clinic, were prospectively screened. The exclusion criteria were as follows: secondary hypertension, hypertensive emergency or urgency, heart failure (New York Heart Association functional class III or VI), ischemic heart or peripheral arterial disease within the previous 6 months, significant arrhythmia, night labor or shift work, pregnancy, history of drug or alcohol abuse within the previous 6 months, and taking drugs known to affect BP, such as steroids, monoamine oxidase inhibitors, oral contraceptives, or sympathomimetics. Three hundred and ninety-nine individuals with valid office, home, and ambulatory BP measurements were finally analyzed. The study protocol was approved by the Institutional Review Board of the hospital, and all of the study subjects provided written informed consent (Clinicaltrial.gov NCT03855605).

### 2.2. Measurements of Office, Home, and Ambulatory BP

The schedule used for the BP measurements was similar to that used in our previous study [7,8]. Briefly, on the first visit day, office BP was measured and instruction for home BP measurements was given to the participants by the study nurse. The home BP measurements started from the evening of the first day, then continued for 7 to 9 consecutive days, and ended in the morning of the 8th to 10th day. On the last day of the home BP measurements, all of the participants visited the clinical trial center of the hospital to measure the second office BP, and to start the 24-h ambulatory BP measurements. On the following day, when the 24-h ambulatory BP measurements had been obtained, the third office BP was measured. During the measurement period, the participants were asked to maintain their daily activities.

The office BP was measured using a validated oscillometric device (WatchBP Office, Microlife, Taiwan) while seated, at both arms, three times, with 1-min intervals, and during the three visits, after a 5-min rest. For the home BP measurements, the participants were instructed to take three measurements at 1-min intervals every morning, after micturition and before breakfast (between 07:00, or waking, and 09:00) and every evening (between 21:00 and 23:00, or bedtime) for 7 to 9 consecutive days, while seated in a quiet place, after 5-min rest, using the provided device (WatchBP Home, Microlife, Taiwan). A valid measurement for a home BP was defined as being at least 5 days of morning and evening measurements. The ambulatory BP monitoring over 24 h was performed on the non-dominant arm using an automated, noninvasive oscillometric device (Mobil-O-Graph, I.E.M. GmbH, Germany) with a measurement interval of 30 min. The participants were instructed to continue their normal daily activities during the ABP measurements. A valid measurement for the ambulatory BP was defined as being valid readings for more than 70% of all of the measurement attempts, and at least 14 measurements during the daytime (09:00 to 21:00) and at least seven measurements during the nighttime (00:00 to 00:60). Blood samples were obtained after an overnight fast of at least 8 h.

### 2.3. Definition of Hypertension

All three morning and evening home BP readings for whole study days [7] and the daytime ambulatory BP readings (09:00 to 21:00 h) were averaged for each participant. The BP thresholds for the diagnosis of hypertension were defined as an average systolic BP (SBP) of ≥135 mmHg, and/or an average diastolic BP (DBP) of ≥85 mmHg for home and daytime ambulatory BP measurements [3]. The classification of the office SBP and DBP (optimal, normal, high normal and hypertension) were determined according to the latest guidelines [3].

### 2.4. Statistical Analysis

The participants were classified into three groups according to their diagnosis of hypertension based on their home BP and ambulatory BP measurements: Agree-NT (home BP and ambulatory BP sustained normotension), Disagree (home BP normotension and ambulatory BP hypertension, or home BP hypertension and ambulatory BP normotension), and Agree-HT (home BP and ambulatory BP sustained hypertension). The BP variability was assessed using the standard deviations (SD) of averaged intra-visit and inter-visit office, home, and ambulatory BP.

The continuous variables were expressed as means ± standard deviations, and categorical variables were expressed as numbers (percentages). The intergroup differences were analyzed using analysis of variance, with post-hoc comparisons for continuous variables and the chi-square test was used for categorical variables, respectively. Multivariate linear regression analysis was used to determine the effects of the independent variables on the difference between home and ambulatory BP. A generalized multinomial logistic regression for three diagnostic categories was applied to identify the predictors of diagnostic disagreement, including independent variables of age, sex, body mass index, smoking status, alcohol drinking, estimated glomerular filtration rate (eGFR) calculated by using the Chronic Kidney Disease Epidemiology Collaboration (CKD-EPI) creatinine equation, diabetes mellitus, office BP, home BP and ambulatory BP. Two-sided *p* values <0.05 were considered statistically significant. The analyses were performed using SPSS version 22.0 (IBM Co., Armonk, NY, USA) and MedCalc Version 19.0 (MedCalc Software Ltd., Ostend, Belgium).

## 3. Results

### 3.1. Participants’ Characteristics

Of the 470 participants, 399 who had valid office, home, and ambulatory BP measurements were finally analyzed and classified into the Agree-NT group (*n* = 56, 14.0%), the Disagree group (*n* = 84, 21.1%), and the Agree-HT group (*n* = 259, 64.9%) (Figure 1). The baseline characteristics of the participants are summarized in Table 1. The participants in the Disagree group were significantly older than those in the Agree-HT group. The proportions of male participants were 26.8%, 42.9%, and 54.4% in the Agree-NT, the Disagree, and the Agree-HT groups, respectively (*p* < 0.001). The creatinine and triglyceride levels of the Disagree group were intermediate between those of the Agree-NT and the Agree-HT groups, with significant differences. The mean electrocardiography Sokolow-Lyon voltage (the voltage of the S wave in V1 + the voltage of the R wave in V5 or V6) of the Disagree group was also between those of the Agree-NT and the Agree-HT groups.

### 3.2. Comparisons of BP among the Groups

Table 2 shows the BP of the three groups. The individuals in the Disagree group had significantly higher office DBP; home DBP; and 24-h, daytime, and nighttime ambulatory BP than those in the Agree-NT group (all *p* < 0.001). However, all of the BP measurements in the Disagree group were significantly lower than those in the Agree-HT group (all *p* < 0.001). Regarding the office BP category, the majority of the Disagree group had a high normal SBP and DBP (51.2% and 33.3%, respectively) or hypertensive SBP and DBP (35.7% and 32.1%, respectively) (Figure 2).

### 3.3. Determinants of the Difference between the Home and Ambulatory BP

In the linear regression analysis, alcohol drinking was significantly associated with an SBP difference between the home and ambulatory BP, and the office BP with a DBP difference, respectively (Table 3). In the generalized multinomial logistic regression analysis for the three diagnostic categories, the variables—except for home and ambulatory BP—showed no significant odds ratio for the Disagree group compared to the Agree-NT or the Agree-HT group (Table 4).

### 3.4. Comparisons of the BP variability among the Groups

The BP variabilities of each group are summarized in Table 5. There were no differences in the intra-visit and inter-visit office BP variabilities among the groups. For the home BP variabilities, the standard deviations of SBP and DBP in the Disagree group were significantly lower than those in the Agree-HT group for the morning and evening measurements, but were not significantly different from those in the Agree-NT group. The standard deviations for 24-h, daytime, and nighttime, and the weighted standard deviation of the 24-h ambulatory BP in the Disagree group were intermediate between those in the Agree-NT and the Agree-HT groups.

### 3.5. Subgroup Analysis of the Disagree Group

In the Disagree group, 17 individuals (20.2%) had home BP hypertension and 67 individuals (79.8%) had ambulatory BP hypertension. The clinical characteristics and laboratory findings were not significantly different between these two groups (Table 6). The office BP was also not different between the groups. However, the home BP values were higher in the home BP hypertension group, and the 24-h, daytime and nighttime ambulatory BP values were higher in the ambulatory BP hypertension group. On the other hand, the nighttime ambulatory BP values were not different between these two groups.

The analysis of the diagnosis based on the 24-h ambulatory BP showed similar results to the analysis of the diagnosis based on the daytime ambulatory BP (data not shown).

## 4. Discussion

The major findings of our study are as follows: (1) the mean office and home DBP in the Disagree group were higher than those in the Agree-NT group; (2) the mean ambulatory SBP and DBP, and the aortic SBP in the Disagree group differed significantly, and were intermediate between those in the Agree-NT and the Agree-HT groups; (3) no consistent clinical or demographic characteristics were identified to be determinants of the BP differences and the diagnostic disagreement between the home and ambulatory BP measurements; and (4) the ambulatory BP variabilities and the ECG voltage of the Disagree group were intermediate between those of the Agree-NT and the Agree-HT groups.

The guidelines recommend out-of-office BP measurements for the optimal diagnosis and treatment of hypertension, because they are more closely associated with cardiovascular outcomes than office BP measurements, and enable white-coat and masked hypertension to be identified [1,3]. Home and ambulatory BP measurements are the representative out-of-office BP measurement methods. Although ambulatory BP measurements are better for cardiovascular risk prediction than home BP measurements [9], the inconvenience and cost are major limitations of using ambulatory BP measurements in clinical practice. Hence, home BP measurements may be used in the diagnosis of hypertension phenotypes. However, the interchangeability of the home and ambulatory BP measurements in the diagnosis of hypertension is controversial because of the diagnostic disagreement between both methods [10,11]. The diagnostic disagreement between the two methods may be problematic in clinical practice because doctors and patients may require treatment decisions based on a clear diagnosis. In addition, the diagnostic disagreement between the home and ambulatory daytime BP measurements is not uncommon, and may lead to misdiagnoses and mistreatments. Therefore, the prediction of patients with diagnostic disagreement, and their risk of cardiovascular disease, would be of great importance in clinical practice.

Few studies have sought to identify the determinants of the diagnostic disagreement. Ntineri et al. found that age, sex, antihypertensive drug treatment, centers, body mass index and office hypertension were associated with the difference between home and daytime ambulatory BP, and antihypertensive drug treatment, alcohol consumption, and office normotension were major determinants of the diagnostic disagreement between home and daytime ambulatory BP measurements [12]. However, we failed to identify any demographic or clinical feature that was capable of distinguishing individuals with diagnostic disagreement between their home and ambulatory BP measurements. A linear regression analysis also failed to identify any factor consistently associated with differences between home and daytime ambulatory BP. Furthermore, the logistic regression analysis showed that home and ambulatory BP were the only significant determinants of diagnostic disagreement between home and ambulatory BP measurements, which suggests that it may be difficult to predict diagnostic disagreement without home and ambulatory BP results. Although ambulatory BP, office and home DBP showed statistical differences in their values according to the groups, there was considerable overlap. The different results between Ntineri’s and our study may be explained by the difference of the study populations. Ntineri’s study included patients on antihypertensive drug treatment (32% of the study population) [12], whereas we excluded patients on antihypertensive drug treatments. Our study results indicate that it is difficult to distinguish patients with a diagnostic disagreement between their home and ambulatory BP measurements using clinical and demographic characteristics.

A diagnostic disagreement based on home and ambulatory BP measurements may partly be explained by a lack of measurement reproducibility and methodological differences. Previous studies have reported standard deviations for test–retest daytime ambulatory BP of 10 to 12 mmHg for SBP and 7 mmHg for DBP [13,14,15], and standard deviations for test–retest home SBP and DBP of 6.9 and 4.7 mmHg, respectively [15]. In another study, the diagnosis of masked hypertension using repeated office, home, and ambulatory BP measurements showed fair-to-moderate reproducibility [16]. In the present study, the home BP was measured between 7 and 9 am, or within 2 h of waking up, and between 9 and 11 pm, or before bedtime, while the daytime ambulatory BP was the averaged BP obtained between 9 am and 9 pm. The BP being measured at different times may also be a cause of the diagnostic disagreement between home and ambulatory BP measurements.

Although we were unable to identify the differentiating features, the intermediate level of the ambulatory BP parameters, the central aortic SBP, and the ambulatory BP variability in the Disagree group being situated between the Agree-NT and the Agree-HT groups suggest that individuals with a diagnostic disagreement between their home and ambulatory BP measurements may also be at an intermediate cardiovascular risk between sustained home and ambulatory normotension and hypertension. Therefore, a diagnostic disagreement between home BP and ambulatory BP measurements may not absolve individuals of cardiovascular risk, and these individuals should not be classified as ‘normotensives’.

We averaged all three readings of the home BP measurements instead of discarding the first reading and averaging the next two readings, because there was no difference in the diagnostic accuracy and reliability between the two methods in our previous study [7].

The present study has a number of limitations that deserve consideration. First, we conducted the analysis using the daytime ambulatory BP, because both the home and daytime ambulatory BP reflect BP during waking hours, and the same BP threshold is used to diagnose hypertension. All or each of the 24-h, daytime, and/or nighttime ambulatory BP can be used in the diagnosis of hypertension. Previous studies have shown that 24-h and nighttime ambulatory BP are more associated with the prognosis than daytime ambulatory BP [4,9,17]. However, the results of the secondary analysis using diagnosis by 24-h ambulatory BP in our study were not different to the primary analysis. Isolated nocturnal hypertension has also been shown to be associated with the risk of cardiovascular events [18], and the inclusion of nighttime BP or isolated nocturnal hypertension in the diagnosis of hypertension may be better than diagnosis based only on the daytime ambulatory BP. The inclusion of isolated nocturnal hypertension in the diagnosis of hypertension would probably increase the diagnostic disagreement rate, because individuals with isolated nocturnal hypertension in our study (n=39) had a higher frequency of normotension diagnosis based on home BP in our study population (76.9%). Additional studies are needed in order to determine the ways in which to interpret a diagnostic disagreement between BP measurements made at different times. Second, the daytime was determined by a fixed time period, not by the individuals’ diaries. Therefore, the home and daytime ambulatory BP measured at different times were used in the diagnosis of hypertension, although both BP were measured during waking hours. Unlike people living in the West, many Koreans eat dinner out, and therefore the measurements of the home BP before dinner that are recommended by European hypertension guidelines [19] are difficult. Third, this cross-sectional study was conducted on individuals suspected to have hypertension. Although the individuals with diagnostic disagreement seemed to have intermediate cardiovascular risk between sustained home and ambulatory normotension and hypertension, their long-term risk should be determined. Fourth, the small sample size of this study may not be enough to evaluate the difference between individuals with home BP hypertension and ambulatory BP hypertension in the Disagree group.

## 5. Conclusions

Individuals showing diagnostic disagreement between home and ambulatory BP measurements may be at an intermediate cardiovascular risk compared with those with sustained home and daytime ambulatory normotension and hypertension. However, the differentiation of the clinical and demographic features of the disagreement between the home and ambulatory BP measurements could not be accomplished. Further research is needed to reduce the diagnostic disagreement between home and ambulatory BP measurements, and to develop methods that can distinguish individuals with diagnostic disagreement.

## Figures and Tables

**Figure 1 healthcare-08-00457-f001:**
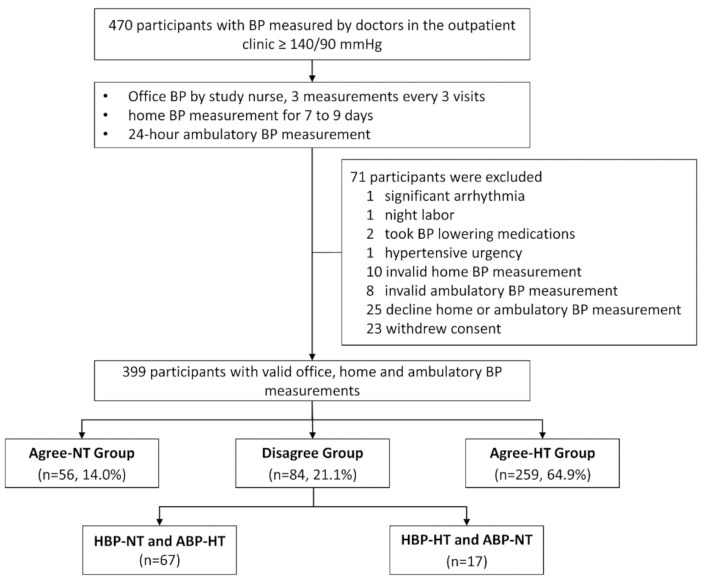
Patients’ dispositions. BP, blood pressure; Agree-NT group, home and daytime ambulatory normotension; Disagree group, home normotension and daytime ambulatory hypertension, or home hypertension and daytime ambulatory normotension; Agree-HT group, home and daytime ambulatory hypertension; HBP-NT, home blood pressure normotension; ABP-HT, ambulatory blood pressure hypertension; HBP-HT, home blood pressure hypertension; ABP-NT, ambulatory blood pressure normotension.

**Figure 2 healthcare-08-00457-f002:**
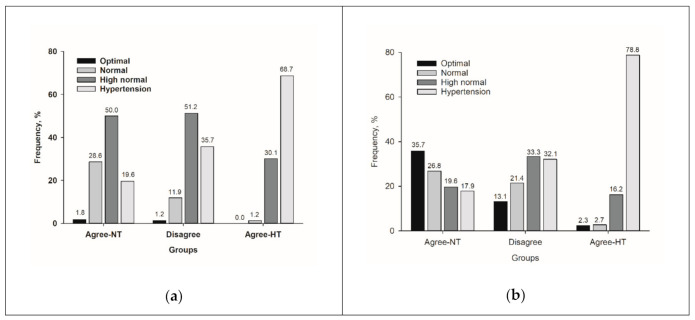
Distributions of office (**a**) systolic and (**b**) diastolic BP classification. Abbreviations: BP, blood pressure; Agree-NT, home and daytime ambulatory normotension; Disagree, home normotension and daytime ambulatory hypertension, or home hypertension and daytime ambulatory normotension; Agree-HT, home and daytime ambulatory hypertension.

**Table 1 healthcare-08-00457-t001:** Baseline clinical characteristics of the study subjects.

	Agree-NT (*N* = 56)	Disagree (*N* = 84)	Agree-HT (*N* = 259)	*p*-Value *
*Clinical characteristics*				
Age, years	56.1 ± 11.0 ^a^	55.9 ± 9.6 ^a^	50.6 ± 9.9 ^b^	<0.001
Sex, male	15 (26.8%)	36 (42.9%)	141 (54.4%)	<0.001
Body mass index, kg/m2	24.7 ± 3.8	25.1 ± 3.5	25.5 ± 3.3	0.292
Diabetes mellitus	2 (3.6%)	9 (10.8%)	22 (8.5%)	0.306
Current smoking	2 (3.6%)	7 (8.3%)	54 (20.8%)	<0.001
*Laboratory findings*				
Creatinine, mg/dL	0.72 ± 0.15 ^a^	0.77 ± 0.20 ^a,b^	0.79 ± 0.17 ^b^	0.017
eGFR	97.2 ± 11.5	95.4 ± 13.9	98.1 ± 13.1	0.250
Glucose, mg/dL	103.3 ± 13.5	103.0 ± 16.6	105.1 ± 24.3	0.677
Hemoglobin A1c, %	5.69 ± 0.44	5.69 ± 0.74	5.73 ± 0.77	0.872
Total cholesterol, mg/dL	204.0 ± 36.1	194.2 ± 33.1	199.2 ± 32.8	0.227
Triglycerides, mg/dL	122.6 ± 55.8 ^a^	132.7 ± 63.4 ^a,b^	158.8 ± 102.4 ^b^	0.006
HDL-cholesterol, mg/dL	57.8 ± 16.1	54.8 ± 13.6	54.3 ± 15.1	0.279
LDL-cholesterol, mg/dL	136.0 ± 32.3	131.4 ± 30.9	132.7 ± 31.0	0.691
Urine albumin-to-creatinine ratio,	20.1 ± 81.0	26.4 ± 102.7	25.3 ± 76.8	0.905
ECG voltage, mV	2.2 ± 0.6 ^a^	2.3 ± 0.7 ^a,b^	2.5 ± 0.7 ^b^	0.007

Data are presented as means ± standard deviation or numbers (percentages). * *p-*value given by the analysis of variance, where the different superscript alphabets (a and b) represent significant differences at an alpha level of 0.05, as given by the post hoc Tukey’s HSD analysis. Abbreviations: HDL-cholesterol, high-density lipoprotein cholesterol; LDL-cholesterol, low-density lipoprotein cholesterol; ECG, electrocardiogram; eGFR, estimated glomerular filtration rate. The eGFR values were calculated using the CKD-EPI creatinine equation. The ECG voltages were calculated using the Sokolow-Lyon voltage (defined as the voltage of the S wave in V1 + the voltage of the R wave in V5 or V6).

**Table 2 healthcare-08-00457-t002:** Comparison of blood pressure.

	Agree-NT (*N* = 56)	Disagree (*N* = 84)	Agree-HT (*N* = 259)	*p*-Value *
*Office BP measurements*				
Office SBP, mmHg	134.1 ± 7.7 ^a^	137.4 ± 7.8 ^a^	145.3 ± 10.2 ^b^	<0.001
Office DBP, mmHg	82.8 ± 6.7 ^a^	87.0 ± 5.6 ^b^	95.6 ± 7.8 ^c^	<0.001
*Home BP measurements*				
Home SBP, mmHg	124.3 ± 6.3 ^a^	127.9 ± 6.5 ^a^	141.4 ± 10.6 ^b^	<0.001
Home DBP, mmHg	76.5 ± 5.7 ^a^	81.4 ± 4.2 ^b^	92.6 ± 7.5 ^c^	<0.001
*Ambulatory BP measurements*				
24-h SBP, mmHg	120.3 ± 5.6 ^a^	127.6 ± 7.0 ^b^	139.9 ± 10.8 ^c^	<0.001
24-h DBP, mmHg	74.4 ± 5.8 ^a^	83.4 ± 5.6 ^b^	93.9 ± 8.6 ^c^	<0.001
Daytime SBP, mmHg	123.5 ± 6.6 ^a^	132.8 ± 7.7 ^b^	144.9 ± 11.6 ^c^	<0.001
Daytime DBP, mmHg	76.8 ± 5.7 ^a^	87.5 ± 6.0 ^b^	98.0 ± 9.2 ^c^	<0.001
Nighttime SBP, mmHg	113.3 ± 8.3 ^a^	118.6 ± 10.5 ^b^	130.4 ± 13.5 ^c^	<0.001
Nighttime DBP, mmHg	69.6 ± 6.6 ^a^	76.0 ± 7.7 ^b^	86.1 ± 10.5 ^c^	<0.001
*Aorta BP measurements*				
Aorta SBP, mmHg	124.1 ± 8.1 ^a^	127.7± 8.3 ^b^	136.3 ± 10.1 ^c^	<0.001

Data are presented as mean ± standard deviation or number (percentage). * *p*-value given by the analysis of variance, where the different superscript alphabets (a, b, and c) represent significant differences at an alpha level of 0.05, as given by the post hoc Tukey’s HSD analysis. Abbreviations: BP, blood pressure; SBP, systolic blood pressure; DBP, diastolic blood pressure.

**Table 3 healthcare-08-00457-t003:** Determinants of the differences between the home and ambulatory blood pressure, as determined by the linear regression analysis.

Variable	Standardized Beta Coefficient	
	Systolic BP Difference	Diastolic BP Difference
Age	0.06	−0.15
Sex	0.07	0.13
BMI	−0.03	−0.07
Smoking	−0.06	−0.07
Alcohol drinking	0.14 *	0.06
Diabetes	−0.07	0.00
eGFR	−0.10	−0.11
Office BP	0.08	−0.14 *
ECG voltage	−0.10	−0.03

* *p* < 0.05 given by the multivariate linear regression analysis, including age, sex, body mass index, smoking status, alcohol drinking, diabetes mellitus, the estimated glomerular filtration rate (eGFR) as given by the CKD-EPI creatinine equation, office blood pressure, and electrocardiographic Sokolow-Lyon voltage. The office systolic blood pressure for the systolic blood pressure difference and the office diastolic blood pressure for the diastolic blood pressure difference were included as covariates. The electrocardiogram (ECG) voltages were calculated using the Sokolow-Lyon voltage (defined as the voltage of the S wave in V1 + the voltage of the R wave in V5 or V6). Abbreviations: BP, blood pressure; BMI, body mass index; eGFR, estimated glomerular filtration rate; ECG, electrocardiogram.

**Table 4 healthcare-08-00457-t004:** Determinants of the diagnostic disagreement between home and ambulatory measurements, as determined by the logistic regression analysis.

Variable	Odd Ratio of the Disagree Group versus the Agree-NT Group		Odd Ratio of the Disagree Group versus the Agree-HT Group	
	Model1	Model2	Model1	Model2
Age	0.98 (0.93–1.03)	0.92 (0.84–1.01)	0.98 (0.94–1.02)	0.99 (0.93–1.06)
Sex	0.62 (0.27–1.45)	0.93 (0.24–3.68)	1.10 (0.57–2.14)	0.88 (0.29–2.67)
BMI	0.95 (0.85–1.07)	1.06 (0.89–1.27)	0.96 (0.88–1.05)	1.11 (0.94–1.30)
Smoking	0.71 (0.13–4.02)	0.48 (0.01–35.52)	2.26 (0.85–6.01)	1.04 (0.27–4.06)
Alcohol drinking	1.05 (0.47–2.33)	0.60 (0.16–2.25)	0.81 (0.43–1.54)	0.83 (0.30–2.27)
eGFR	1.01 (0.98–1.05)	0.99 (0.93–1.06)	1.00 (0.97–1.02)	1.00 (0.97–1.04)
Office SBP	0.99 (0.93–1.04)	1.00 (0.89–1.13)	1.07 (1.02–1.11)	1.06 (0.96–1.16)
Office DBP	0.90 (0.83–0.97)	1.01 (0.85–1.20)	1.16 (1.09–1.24)	1.05 (0.91–1.21)
Diabetes	0.36 (0.07–1.86)	0.31 (0.03–3.45)	0.98 (0.37–2.58)	0.77 (0.16–3.65)
Home SBP		0.90 (0.78–1.04)		1.27 (1.13–1.43) *
Home DBP		0.84 (0.67–1.05)		1.46 (1.23–1.73) *
Daytime SBP		0.85 (0.75–0.95) *		1.06 (0.98–1.15)
Daytime DBP		0.70 (0.59–0.83) *		1.16 (1.02–1.31) *

* *p* < 0.05 given by the generalized multinomial logistic regression analysis. Abbreviations: BP, blood pressure; SBP, systolic BP; DBP, diastolic BP; BMI, body mass index; eGFR, estimated glomerular filtration rate as given by the CKD-EPI creatinine equation.

**Table 5 healthcare-08-00457-t005:** Comparisons of the blood pressure variability.

BP Variability	Agree-NT (*N* = 56)	Disagree (*N* = 84)	Agree-HT (*N* = 259)	*p*-Value *
*Office BP variability*				
Standard deviation of SBP, intra-visit, mmHg	4.86 ± 1.80	4.73 ± 1.80	4.63 ± 1.77	0.665
Standard deviation of DBP, intra-visit, mmHg	2.73 ± 1.26	2.62 ± 1.12	2.86 ± 1.22	0.259
Standard deviation of SBP, inter-visit, mmHg	7.57 ± 3.82	7.67 ± 4.28	7.18 ± 4.53	0.611
Standard deviation of DBP, inter-visit, mmHg	4.65 ± 2.65	4.25 ± 2.53	4.54 ± 2.45	0.565
*Home BP variability*				
Standard deviation of SBP, mmHg	8.57 ± 2.30 ^a^	8.43 ± 2.00 ^a^	9.74 ± 2.82 ^b^	<0.001
Standard deviation of DBP, mmHg	5.23 ± 1.38 ^a^	5.41 ± 1.53 ^a^	6.19 ± 2.46 ^b^	0.001
Standard deviation of SBP, morning, mmHg	7.20 ± 2.58 ^a^	7.13 ± 2.02 ^a^	8.14 ± 2.60 ^b^	0.001
Standard deviation of DBP, morning, mmHg	4.39 ± 1.66	4.69 ± 1.82	5.07 ± 2.12	0.043
Standard deviation of SBP, evening, mmHg	8.28 ± 2.60 ^a^	8.49 ± 2.52 ^a^	9.77 ± 3.51 ^b^	<0.001
Standard deviation of DBP, evening, mmHg	5.16 ± 1.71 ^a^	5.38 ± 1.53 ^a^	6.40 ± 3.26 ^b^	0.001
*Ambulatory BP variability*				
Standard deviation of 24-h SBP, mmHg	13.4 ± 3.7 ^a^	14.3 ± 3.7 ^a,b^	15.6 ± 4.6 ^b^	0.001
Standard deviation of 24-h DBP, mmHg	9.5 ± 2.4 ^a^	11.1 ± 2.5 ^b^	12.0 ± 3.0 ^c^	<0.001
Standard deviation of daytime SBP, mmHg	12.6 ± 4.0	12.5 ± 4.4	13.7 ± 4.9	0.043
Standard deviation of daytime DBP, mmHg	8.9 ± 2.7 ^a^	9.4 ± 3.1 ^a,b^	10.1 ± 3.2 ^b^	0.017
Standard deviation of nighttime SBP, mmHg	9.9 ± 4.3 ^a^	10.4 ± 3.9 ^a,b^	11.5 ± 4.4 ^b^	0.012
Standard deviation of nighttime DBP, mmHg	7.7 ± 2.7 ^a^	8.4 ± 2.9 ^a^	9.6 ± 3.3 ^b^	<0.001
Weighted standard deviation of 24-h SBP, mmHg	11.7 ± 3.5 ^a^	11.8 ± 3.3 ^a,b^	13.0 ± 3.8 ^b^	0.005
Weighted standard deviation of 24-h DBP, mmHg	8.5 ± 2.3 ^a^	9.1 ± 2.4 ^a,b^	9.9 ± 2.6 ^b^	<0.001

Data are presented as means ± standard deviation or numbers (percentages). * *p-*value given by the analysis of variance, where the different superscript alphabets (a, b, and c) represent significant differences at an alpha level of 0.05, as given by the post hoc Tukey’s HSD analysis. Abbreviations: BP, blood pressure; SBP, systolic blood pressure; DBP, diastolic blood pressure.

**Table 6 healthcare-08-00457-t006:** Demographic and clinical characteristics of the individuals in the Disagree group.

	HBP Hypertension (*N* = 17)	ABP Hypertension (*N* = 67)	*p*-Value
*Clinical characteristics*			
Age, years	58.8 ± 10.4	54.5 ± 8.2	0.073
Male	8 (47.1%)	28 (41.8%)	0.695
Body mass index, kg/m^2^	25.0 ± 3.9	25.1 ± 3.5	0.916
Diabetes mellitus	2 (11.8%)	7 (10.6%)	0.891
*Laboratory findings*			
Creatinine, mg/dL	0.81 ± 0.24	0.76 ± 0.19	0.351
Glucose, mg/dL	103.2 ± 11.0	103.0 ± 17.9	0.967
Hemoglobin A1c, %	5.66 ± 0.40	5.70 ± 0.81	0.853
Total cholesterol, mg/dL	194.1 ± 33.6	194.3 ± 33.2	0.983
Triglycerides, mg/dL	124.7 ± 69.5	134.8 ± 62.1	0.558
HDL-cholesterol, mg/dL	50.8 ± 15.9	55.9 ± 12.9	0.176
LDL-cholesterol, mg/dL	134.5 ± 33.3	130.6 ± 30.5	0.647
Urine albumin-to-creatinine ratio	22.8 ± 55.1	26.5 ± 112.4	0.896
ECG voltage, mm	24.2 ± 10.5	22.8 ± 6.5	0.504
*BP measurements*			
Office SBP, mmHg	139.0 ± 7.3	137.0 ± 7.9	0.356
Office DBP, mmHg	86.6 ± 5.8	87.1 ± 5.6	0.747
Home SBP, mmHg	134.7 ± 4.8	126.1 ± 5.7	<0.001
Home DBP, mmHg	85.2 ± 5.6	80.5 ± 3.1	0.004
24-h SBP, mmHg	123.6 ± 5.8	128.6 ± 6.9	0.007
24-h DBP, mmHg	77.8 ± 4.9	84.9 ± 4.8	<0.001
Daytime SBP, mmHg	125.7 ± 4.7	134.6 ± 7.3	<0.001
Daytime DBP, mmHg	77.8 ± 4.9	84.9 ± 4.8	<0.001
Nighttime SBP, mmHg	118.2 ± 9.1	118.7 ± 10.8	0.868
Nighttime DBP, mmHg	73.1 ± 8.5	76.8 ± 7.3	0.080
Aorta SBP, mmHg	129.2 ± 7.2	127.4 ± 8.5	0.431

Data are presented as means ± standard deviation, or numbers (percentages). The ECG voltages were calculated using the Sokolow-Lyon voltage (defined as the voltage of the S wave in V1 + the voltage of the R wave in V5 or V6). Abbreviations: BP, blood pressure; HDL-cholesterol, high-density lipoprotein cholesterol; LDL-cholesterol, low-density lipoprotein cholesterol; ECG, electrocardiogram; SBP, systolic blood pressure; DBP, diastolic blood pressure.

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
