# Peer review of "Characteristics of Individuals with Disagreement between Home and Ambulatory Blood Pressure Measurements for the Diagnosis of Hypertension"

_healthcare, 2020, doi:10.3390/healthcare8040457_

Round 1

Reviewer 2 Report

The manuscript presents data from an observational, prospective cohort clinical trial investigating blood pressure measurements  protocols (ambulatory vs home). Since the consensus still hasn't been reached regarding the appropriate and most accurate BP measurement protocols for normotensive, hypertensive or individuals with white coat syndrome, studies like the one presented in this manuscript are imperative to be completed and the authors are commended for that.

My recommendation is to accept the manuscript after considerations by the authors of the following two points:

  1. Please indicate which formula was used for eGFR measurements.
  2. Please discuss briefly why all three home BP measurements were averaged as opposed to only the latter two.

A minor typographical error is in the spelling of the LVH criteria index: should be Sokolow Lyon voltage rather than Sokolov Lyon.

Round 2

Reviewer 1 Report

The authors have addressed my concerns